# Molecular Signature of Long Non-Coding RNA Associated with Areca Nut-Induced Head and Neck Cancer

**DOI:** 10.3390/cells12060873

**Published:** 2023-03-11

**Authors:** Hung-Han Huang, Guo-Rung You, Shang-Ju Tang, Joseph T. Chang, Ann-Joy Cheng

**Affiliations:** 1Graduate Institute of Biomedical Sciences, College of Medicine, Chang Gung University, Taoyuan 33302, Taiwan; 2Department of Medical Biotechnology and Laboratory Science, College of Medicine, Chang Gung University, Taoyuan 33302, Taiwan; 3Department of Radiation Oncology and Proton Therapy Center, Linkou Chang Gung Memorial Hospital, Taoyuan 333423, Taiwan; 4School of Medicine, Chang Gung University, Taoyuan 33302, Taiwan

**Keywords:** head and neck cancer (HNC), long non-coding RNA (lncRNA), areca nut, clinical significance, cell invasion, UCA1, LUCAT1, MIR31HG

## Abstract

The areca nut is a high-risk carcinogen for head and neck cancer (HNC) patients in Southeast Asia. The underlying molecular mechanism of areca nut-induced HNC remains unclear, especially regarding the role of long non-coding RNA (lncRNA). This study employed a systemic strategy to identify lncRNA signatures related to areca nut-induced HNC. In total, 84 cancer-related lncRNAs were identified. Using a PCR array method, 28 lncRNAs were identified as being dysregulated in HNC cells treated with areca nut (17 upregulated and 11 downregulated). Using bioinformatics analysis of The Cancer Genome Atlas Head-Neck Squamous Cell Carcinoma (TCGA-HNSC) dataset, 45 lncRNAs were differentially expressed in tumor tissues from HNC patients (39 over- and 6 under-expressions). The integrated evaluation showed 10 lncRNAs dysregulated by the areca nut and altered expression in patients, suggesting that these panel molecules participate in areca nut-induced HNC. Five oncogenic (LUCAT1, MIR31HG, UCA1, HIF1A-AS2, and SUMO1P3) and tumor-suppressive (LINC00312) lncRNAs were independently validated, and three key molecules were further examined. Pathway prediction revealed that LUCAT1, UCA1, and MIR31HG modulate multiple oncogenic mechanisms, including stress response and cellular motility. Clinical assessment showed that these lncRNAs exhibited biomarker potentials in diagnosis (area under the curve = 0.815 for LUCAT1) and a worse prognosis (both *p* < 0.05, survival analysis). Cellular studies further demonstrated that MIR31HG facilitates areca nut-induced cancer progression, as silencing this molecule attenuated arecoline-induced invasion ability in HNC cells. This study identified lncRNA signatures that play a role in areca nut-induced HNC. These molecules may be further applied in risk assessment, diagnosis, prognosis, and therapeutics for areca nut-associated malignancies.

## 1. Introduction

Head and neck cancer (HNC) is one of the most common cancers worldwide, predominantly occurring in males [1]. Most HNC cases are squamous cell carcinomas and often arise from mucosal surfaces of major sites, including the oral cavity, hypopharynx, larynx, nasopharynx, and sphenoid sinus [2]. The five-year survival rate for early oral cancer exceeds 80% while decreasing by over two-fold at the mid-to-late stage. This poor prognosis may be due to the rich lymph node tissues at the neck. Patients diagnosed at an advanced stage often present with high tumor heterogeneity and regional neck lymph node invasion [3]. The standard treatment methods for HNC include surgery, radiotherapy, and chemotherapy alone or in combination. Radiotherapy aims to deliver a specific loco-regional radiation dose to destroy tumors while limiting the exposure of surrounding healthy tissues. Recent advancements in radio-therapeutic modality have been demonstrated to improve patient survival or life quality. For example, proton beam radiotherapy treating HNC has shown a higher benefit than conventional X-ray photo radiotherapy [4]. Chemotherapy is considered a systemic treatment that may affect the whole body to eradicate the spreading of cancer cells. The innovative chemotherapeutic approach has expanded the primary goal of cancer cure to adjuvant sensitization, palliation, or prevention. For example, several herbs or bio-drugs have been reported to reduce cancer risk or potential adverse side effects [5]. Other novel approaches, such as sonodynamic therapy, have also shown promise as an alternative treatment for HNC [6]. The association between potential carcinogen exposure and HNC has been reported previously. While cigarette smoking and alcohol consumption are more closely related to the disease in Western countries, areca nut chewing is a common habit for HNC patients in Southeast Asia. Patients with combined habits of alcohol consumption, smoking, and areca nut chewing have the highest mortality [7,8]. Globally, over 600 million people use areca nuts or betel quids, making it the fourth most common habitual substance after tobacco, alcohol, and caffeine [9]. Arecoline, a highly abundant alkaloid in areca nuts, is the most active ingredient associated with pathological development [9,10]. The areca nut and arecoline are genotoxic and cytotoxic in HNC cells via several mechanisms, such as reactive oxygen species generation, DNA damage, tissue hypoxia, cell migration, and invasion [9,10,11,12]. Recent studies using chronic exposure models have revealed comprehensive pathological effects. These include anchorage-independent growth, epithelial-mesenchymal transition (EMT), and cancer stemness conversion, which may lead to chemo-radioresistance or distant metastasis in HNC [12,13,14]. HNC patients with an areca nut chewing habit often exhibit more aggressive attributes, including higher incidences of second primary tumors, residual microsatellite tumors, and poor survival [14,15,16]. At the molecular level, numerous altered expression and dysregulated pathways have been found to be related to specific malignant phenotypes [9,10,11,12,13,14]. However, how areca nuts contribute to carcinogenesis is still unclear.

Useful cancer biomarkers should significantly improve the efficacy of disease management. Clinically, cancer biomarkers can be applied in cancer screening, diagnosis, prognosis, monitoring disease progression, predicting therapeutic response, or risk assessment [17]. Examples are the markers of cancer stem cells in oral cancer, which are associated with the pathological stem-like features of recurrence and metastasis that may predict chemo-drug resistance and treatment efficacy [18]. As clinical samples, biomarkers can be detected in tissues (biopsy) or circulating fluids (liquid biopsy), such as serum/plasma or saliva. The advantage of tissue biopsy is precisely correlating tumor status in situ, while liquid biopsy presents an overall tumor heterogeneity and involves minimally invasive procedures [19]. At the molecular level, biomarkers can be categorized as DNA, RNA, protein, or other metabolites. For over a decade, the circulating nucleic acids have been considered valuable biomarkers, for they may be highly elevated in cancer patients [19]. An example is the miR-196b as a promising salivary cancer marker to detect early HNC and oral precancer lesions with high sensitivity and specificity [20]. Nevertheless, the role of the novel biomolecular family, the long non-coding RNA (lncRNA), in HNC has yet to be addressed.

It is well established that while more than 75% of the human genome is transcribed, less than 3% consists of coding genes [21,22]. The remaining non-coding RNAs (ncRNAs) have recently become increasingly recognized. They can be divided into two classes according to their transcript size: small ncRNAs (<200 nucleotides) and long ncRNAs (lncRNAs; >200 nucleotides). The many types of small ncRNAs include microRNAs (miRNAs), small nuclear RNA, transfer RNA, and circular RNA. The lncRNA family can be categorized into several biotypes: sense, antisense, intergenic, bidirectional, and intronic lncRNAs, based on their location in the protein-coding genes in the genome [21,22]. The broad range of regulatory mechanisms of lncRNAs involves unique interactions with DNA, RNA, or protein molecules. For example, lncRNAs can recruit different chromatin remodeling proteins to alter their chromatin organizational patterns. They can act as ‘sponges’ by pairing with complementary miRNAs to reduce their effects. LncRNAs can play scaffolding roles by providing docking sites for proteins that function together in the same biological pathway. They can modulate mRNA function through base pairing and inhibiting translation, altering splicing patterns, and subjecting them to degradative pathways [22,23]. Recently, lncRNAs were found to participate in tumorigenesis via multiple regulatory pathways. These include cell proliferation, invasion, metabolic disorders, immune escape, and the maintenance of cancer stemness [23,24,25].

The Cancer Genome Atlas (TCGA) provides a comprehensive genomic library for multiple cancers, including head-neck squamous carcinoma (TCGA-HNSC). Based on this dataset, several lncRNA profiles associated with specific carcinogenic attributes were constructed. These include the effects of alcohol consumption [26], human papillomavirus infection [27], immunity or tumor microenvironment [28,29], and their association with prognosis [29,30,31,32,33]. However, most studies solely analyzed big data without further experimental validation, which may limit the pathological insight of candidate molecules. On the other hand, although the areca nut is a primary carcinogen of HNC in Southeast Asia, the knowledge of lncRNA in areca nut-associated carcinogenesis is very limited. To the best of our knowledge, only one report has mentioned the relevance of lncRNAs in areca nut-induced HNC. Li et al. performed a gene chip analysis of lncRNAs that were differentially expressed between the tumor and the adjacent normal tissues from tongue cancer patients with an areca nut chewing habit [34]. Although thousands of lncRNAs may establish potential gene regulatory networks, no specific molecules have been identified. A more extensive investigation of lncRNAs in the carcinogenic function of areca nuts leading to HNC is warranted.

This study employed a systemic strategy to identify lncRNA signatures that may confer to areca nut-induced HNC, aiming to provide a knowledge foundation for further applications in risk assessment, diagnosis, prognosis, or therapeutic targets. We analyzed 84 cancer-related lncRNAs that may respond to treatment with an areca nut extract (ANE). We assessed the clinical relevance of these 84 molecules using TCGA-HNSC dataset. The integrated lncRNAs were further confirmed to identify molecular signatures induced by areca nuts to facilitate HNC. We further explored the molecular pathways and performed cellular studies on three hub lncRNAs (LUCAT1, UCA1, and MIR31HG) to demonstrate their roles in modulating the areca nut-induced malignant function. Our results provide valuable pathological insights into areca nut-induced malignancy.

## 2. Materials and Methods

### 2.1. Cells, Cell Culture, and Areca Nut Treatment

HNC cell lines OECM1, SAS, FaDu, Detroit 562 (Detroit), and CGHNC8 were used in this study. The specific information of each cell line, including the accession number of the database, is described in Supplementary Cell line information. The cell culture conditions were the same as those previously described [35,36]. Briefly, OECM1 cells were grown in RPMI 1640 medium (Thermo Fisher Scientific, Waltham, MA, USA), FaDu and Detroit cells were cultured in MEM medium (Thermo Fisher Scientific), and SAS and CGHNC8 cells were maintained in DMEM medium (Thermo Fisher Scientific). The culture media were supplemented with 10% fetal bovine serum (FBS) and 1% antibiotic antimycotic. All cell lines were maintained at 37 °C in a humidified atmosphere containing 5% CO_2_ air.

To establish isogenic sublines of HNC cells with chronic areca nut exposure, each cell line was chronically treated with an IC30 dose (30% cytotoxicity dose) of ANE in a complete culture medium for 3 months, as previously described [12,37]. This IC30 dose of arecoline with short-term treatment (24 h) in various HNC cell lines ranged approximately from 100 to 200 μM (Appendix A). To determine the effect of lncRNA in the regulation of area nut-modulated cellular function, HNC cells were exposed to 200 μM arecoline in a complete culture medium for 24 h, following molecular and functional analyses.

### 2.2. Screening of lncRNAs Using an RT-qPCR Array

Three HNC cell lines (OECM1, SAS, and FaDu) and their respective sublines treated with chronic ANE were used for lncRNA screening. The Human Cancer PathwayFinder RT2 PCR array system consisting of 84 cancer-related lncRNA assays was used for lncRNA screening (QIAGEN GmbH, Hilden, Germany). The lncRNA PCR array was performed according to the manufacturer’s protocol as previously described [33]. Briefly, cell pellets were harvested, washed with PBS, and subjected to RNA isolation using TRIzol reagent (Life Technologies, Carlsbad, CA, USA). After the cDNA was converted, 84 qPCR reactions were performed using each specific lncRNA primer set in parallel in one run in a 96-well plate. The qPCR results were output and analyzed using the GeneGlobe Data Analysis Center. The lncRNAs showing more than 1.5-fold altered expression in the ANE sublines compared to the parental cells were selected as the areca nut responsive molecules.

### 2.3. Analysis of lncRNAs by RT-qPCR

The expression level of each lncRNA was analyzed by the RT-qPCR as previously described [33]. Briefly, total RNA was extracted using the TRIzol reagent (Life Technologies). cDNA was synthesized by incubating RNA in a buffer mixture containing random hexamers and reverse transcriptase (Bionovas, Toronto, ON, Canada). qPCR was performed using the TaqMan qPCR assay system with SYBR Green Supermix reagents (Bio-Rad, Hercules, CA, USA), cDNA, and forward and reverse primers. The specific primers used to analyze lncRNAs are listed in Appendix A. All lncRNA transcription levels were normalized to GAPDH as an internal control.

### 2.4. Knockdown of MIR31HG Expression via Small Interfering RNA (siRNA) Transfection

MIR31HG knockdown was accomplished using specific siRNAs (Invitrogen, Carlsbad, CA, USA). The siRNA sequences are listed in Appendix A. HNC cells were transfected with 300 pmol of each specific siRNA using Lipofectamine 2000 (Invitrogen, Carlsbad, CA, USA) in Opti-MEM media (Invitrogen, Carlsbad, CA, USA), according to the manufacturer’s instructions. After 24 h, Opti-MEM media was replaced with a complete medium. The cell pellet was collected, and the functional assay was performed as follows.

### 2.5. Cellular Invasion Assay

The cell invasion assay was performed as previously described [35,36,37]. Briefly, 5% Matrigel (Becton Dickinson Biosciences, Bedford, MA, USA) was used to coat the membrane of the upper insert of a Millicell invasion chamber (Millipore, Burlington, MA, USA) with a pore size of 8 μm in a 24-well plate. Cells in a medium containing 1% FBS were seeded into the upper insert. The lower chamber contained a medium with 20% FBS to trap invading cells. After 16 to 24 h of incubation at 37 °C, the invasive cells were determined by observing the reverse side of the upper insert after being fixed with formaldehyde and stained with crystal violet.

### 2.6. Public Data, Bioinformatics, and Statistical Analyses

Public RNA sequencing data from TCGA-HNSC dataset were downloaded from the University of California, Santa Cruz (UCSC) Cancer Genome Browser [38]. Data processing and statistical analyses were performed using GraphPad Prism 9.0 (GraphPad Software, San Diego, CA, USA). This dataset contained 546 samples from patients with HNC, with 500 primary tumors, two metastatic tumors, and 44 adjacent normal tissues. We used data from 43 paired samples to investigate the expression of lncRNAs and mRNA. We used data from 500 HNC tumors with comprehensive clinical information to analyze the association between lncRNA expression and survival status.

Spearman correlation analysis was used to determine the co-expression status between mRNAs (genes) and a specific lncRNA using 43 TCGA-HNSC paired samples. Genes with correlation coefficients r > 0.3 and *p* < 0.05 were filtered out as potential lncRNA effectors. To explore lncRNA regulatory mechanisms, we uploaded the lncRNA effector genes to the Database for Annotation, Visualization, and Integrated Discovery website (https://david.ncifcrf.gov/tools.jsp, accessed on 27 June 2022), followed by an analysis of the regulatory pathways based on the Kyoto Encyclopedia of Genes and Genomes (KEGG) dataset.

To evaluate the diagnostic efficacy of each lncRNA, we performed receiver operating characteristic (ROC) analysis to differentiate between normal and tumor samples. We measured the area under the curve (AUC) to evaluate the discriminative power of lncRNAs. Clinical sensitivity and specificity were determined using Youden’s index analysis (sensitivity + specificity −1 is maximal) to obtain an optimal cutoff threshold for maximum diagnostic effectiveness. To evaluate the prognostic significance of each lncRNA, we used Kaplan-Meier analysis to determine the association between the survival status and lncRNA expression levels. High- and low-risk groups were classified using an optimization algorithm in the order of the prognostic index according to the lncRNA expression level. The hazard ratios (HRs) with 95% confidence intervals were calculated using the log-rank tests. Statistical significance was set at *p* < 0.05. 

## 3. Results

### 3.1. Identification of lncRNAs Induced by Areca Nuts

To explore potential lncRNAs in HNC cells that may respond to areca nuts, we analyzed the differential levels of 84 cancer-related lncRNAs in three HNC cell lines that were untreated or treated with ANE. With a dysregulation level exceeding 1.5-fold as the selection criterion, a total of 31, 21, and 38 lncRNAs were found in OECM1, SAS, and FaDu cells, respectively (Figure 1a). The most significantly dysregulated molecules in each cell line are shown in Figure 1b. To obtain a common molecular portrait, lncRNAs that were altered in more than one cell line were selected. A total of 28 lncRNAs were filtered out, with 17 upregulated and 11 downregulated in areca nut-treated cells (Table 1). The overall view of the dysregulation levels with the average fold-change and statistical *p*-values for each lncRNA is shown in Figure 1c. LUCAT1 and KRASP1 displayed the highest elevations, while H19 and LNC00312 were the most inhibited in areca nut-treated HNC cells. The findings defined a lncRNA panel with 28 molecules induced by areca nuts.

### 3.2. Dysregulated lncRNAs in HNC Patients

To determine the clinical significance of the 84 lncRNA molecules in HNC, we assessed the differential expression levels between normal (N = 44) and tumor tissues (N = 497) from HNC patients in TCGA-HNSC dataset. With *p* < 0.05 as the selection criterion, the expressions of 45 molecules were significantly altered (Figure 2a). Figure 2b shows the overall portrait of these molecules in each tissue, with overexpression of 39 lncRNAs and under-expression of six in the tumor tissues (Table 2). Figure 2c presents a few examples of overexpressed molecules that include LUCAT1, MIR31HG, UCA1, HIF1A-AS2, and SUMO1P3. Figure 2d shows the underexpressed lncRNAs: IPW, LINC00312, NAMA, H19, and CBR3-AS1. Thus, we defined a lncRNA panel of 45 molecules dysregulated in patients with HNC.

### 3.3. LncRNA Signature Associated with Areca Nut-Induced HNC

To identify lncRNAs that may contribute to areca nut-induced HNC, we analyzed two lncRNA panels in combination. One panel was comprised of the defined 28 molecules in response to areca nut treatment. The other panel was comprised of the 45 molecules altered in HNC patients. Ten molecules were filtered out with the selection criteria of average |fold regulation| > 1.5 in the ANE treatment panel and *p* < 0.05 in TCGA-HNSC dataset (Figure 3a). These combined lncRNAs included eight oncogenic features: upregulated by areca nut and overexpressed in tumors, and two with tumor-suppressive behaviors: downregulated by areca nut and underexpressed in tumors.

RT-qPCR was performed independently to confirm the lncRNA screening results. The dysregulation level of each lncRNA in response to areca nut was assessed by treating five HNC cell lines with arecoline. The results are shown in Figure 3b. Although various levels were evident in different cell lines, five molecules (LUCAT1, MIR31HG, UCA1, HIF1A-AS2, and SUMO1P3) were consistently upregulated by arecoline, whereas LINC00132 was significantly downregulated. Three lncRNAs (LUCAT1, MIR31HG, UCA1) were further examined for the effect of arecoline by treating cells with serial doses (0~200 μM) or various time points (0~72 h) in three HNC cell lines (OECM1, FaDu, and Detroit). In general, these molecules exhibited up-regulation by arecoline in the dose-dependent (Figure 3c) and time course-dependent (Figure 3d) manners. These results confirmed the genuineness of the screening results. The differential expression of each lncRNA was examined in 12 HNC cell lines and 10 normal oral keratinocytes. The results are shown in Figure 3e. The five lncRNAs upregulated by arecoline were significantly overexpressed in HNC cell lines, while LINC00312, which was downregulated by arecoline, was underexpressed in HNC cells. The findings identify a panel of lncRNA signatures with five oncogenic and one tumor-suppressive features that may contribute to areca nut-induced HNC.

### 3.4. LUCAT1, UCA1, and MIR31HG lncRNAs Have Diverse Oncogenic Functions

Three oncogenic lncRNAs were confirmed based on the highest arecoline-mediated upregulation (LUCAT1 and MIR31HG) or the most significant overexpression in HNC cell lines (UCA1). They were selected for further investigation. To explore the potential mechanism of the lncRNAs in HNC, the correlative expression of genes as candidate effectors of lncRNA modulation was explored. TCGA-HNSC data of comprehensive lncRNA and mRNA information from 43 HNC patients with paired tumor and normal samples were downloaded and evaluated. Figure 4a shows a strategic search scheme for lncRNA functional pathways in HNC. The correlative expression between each specific lncRNA and mRNA was calculated using the Spearman statistical method. With the selection criteria of correlation coefficient r > 0.3 and the *p* < 0.05, a total of 1607, 910, and 1547 mRNAs were positively associated with LUCAT1, UCA1, and MIR31HG, respectively. These mRNAs were imported into the KEGG database to determine the potential participation of functional pathways for each lncRNA. 

The LUCAT1 expression results are shown in Figure 4b. DNA damage repair functions (including base excision and mismatch repair) and stress-associated responses (such as cell cycle, cellular senescence, and apoptosis) were apparent. A motility-related mechanism involving the actin cytoskeleton was also identified. Examples of correlative molecules in these pathways are HMGB1 (r = 0.411), CDC25C (r = 0.453), PNN (r = 0.444), and ENAH (r = 0.382). 

The results for UCA1 are shown in Figure 4c. The mechanisms associated with cellular motility were obvious. These include adherent junctions, tight junctions, extracellular matrix receptor interactions, proteoglycans, and focal adhesions. In addition, other mechanisms related to transcriptional efficacy or mRNA maturation include proteasomes, nucleocytoplasmic transport, and mRNA surveillance. Examples of correlative molecules are PSMD7 (r = 0.402), AMOTL2 (r = 0.490), ORC3 (r = 0.445), and LAMAB3 (r = 0.349).

The results for MIR31HG are shown in Figure 4d. Functional pathways related to cell proliferation (such as cell cycle regulation) and cell motility (such as focal adhesion, regulation of actin cytoskeleton, adherent junctions, and tight junctions) were evident. Furthermore, the participation of MIR31HG in multiple molecular signaling pathways, including MAP-kinase, PI3K-AKT, mTOR, Ras, and Rap1, indicates that MIR31HG modulates a wide range of homeostatic and oncogenic functions. Examples of positively correlated molecules are ITGB4 (r = 0.736), PXN (r = 0.638), FLNB (r = 0.522), and PIK3CD (r = 0.544).

### 3.5. LUCAT1, UCA1, and MIR31HG Are Overexpressed in Tumors and Are Associated with Poor Prognosis

To determine the clinical significance of LUCAT1, UCA1, and MIR31HG in HNC, we evaluated the associations of molecular expression with clinical presentations to assess their predictive power in diagnosis and prognosis using TCGA-HNSC data of the 43 HNC patients with paired normal and tumor samples. As shown in Figure 5a, LUCAT1, UCA1, and MIR31HG were significantly increased in the cancer group (*p* < 0.05), particularly LUCAT1 and MIR31HG (*p* < 0.0001). 

ROC analysis was performed to evaluate the diagnostic power of these molecules using complete TCGA-HNSC dataset. The results are shown in Figure 5b. LUCAT1, UCA1, and MIR31HG yielded an AUC of 0.765, 0.532, and 0.682, respectively. When applying Youden’s index (sensitivity + specificity −1 is maximal) to obtain the optimal cutoff for diagnosis, the respective clinical sensitivity and specificity values were 78% and 66% for LUCAT1, 19% and 100% for UCA1, and 57% and 79% for MIR31HG (Table 3). Since LUCAT1 can distinguish between normal and cancer groups with high efficacy, it may serve as an excellent diagnostic biomarker for HNC. 

To assess the prognostic power of LUCAT1, UCA1, and MIR31HG, we examined the association of their expression levels with patient survival in TCGA-HNSC cohort patients. The results are shown in Figure 5c. High levels of these molecules were all statistically associated with poor survival in HNC patients (HR = 1.368, 1.640, and 1.423, respectively, for LUCAT1, UCA1, and MIR31HG) (all *p* < 0.05). UCA1 was even more promising in prognosis. These lncRNAs may serve as prognostic biomarkers to predict unfavorable treatment outcomes in patients with HNC.

### 3.6. MIR31HG Regulates Areca Nut-Induced Cell Invasion in HNC Cells

The bioinformatics analysis reveals MIR31HG participates in a broader range of molecular signaling pathways related to oncogenic mechanisms (Figure 4), suggesting this molecule plays a crucial role in malignant transformation. Therefore, MIR31HG was selected for further functional confirmation. Cell invasion is a prominent areca nut-facilitating attribute [10,11,12,13] and was commonly revealed by all three lncRNAs (Figure 4). Thus, cell invasion was further investigated. Matrigel invasion assays were performed in the OECM1, CGHNC8, and SAS HNC cell lines after knockdown of expression by MIR31HG-specific siRNA. As shown in Figure 6a, similar effects were evident in the three cell lines that silencing MIR31HG inhibited cell invasion. Areca nuts promote this invasive effect. However, areca nut-induced cell invasion was abolished by MIR31HG silencing. To extend the investigation to the molecular level, we examined the expressions of the genes associated with cell migration or invasion in response to MIR31HG silencing. These genes included the dynamic cytoskeleton molecules filamin B (FLNB), laminin A3 (LAMA3), and metalloproteinase MMP1. As shown in Figure 6b, although various levels across HNC cell lines, these molecules were down-regulated upon MIR31HG silencing. These results supported the cellular function of MIR31HG and inferred that MIR31HG knockdown reduced cytoskeleton turnover and the MMP activity in HNC. We further evaluated the expressions of these genes in cancer tissues using TCGA-HNSC dataset. As shown in Figure 6c, all these genes were significantly overexpressed in tumor tissues (*p* < 0.001). The expressions in the tumor tissues also generally correlated with MIR31HG levels (r = 0.523, r = 0.367, and r = 0.287 for FLNB, LAMA3, and MMP1, respectively). These results demonstrate that MIR31HG regulates the areca nut-induced invasion of HNC cells. 

## 4. Discussion

With advancements in molecular biology, the important roles of lncRNAs in tumorigenesis have been recognized. Although various studies have profiled lncRNAs across different cancers, very few have focused on the intricacies of lncRNAs in HNC. Additionally, knowledge related to carcinogenesis by areca nut, the primary habitual carcinogen used in Southeast Asia, is limited. This study employed a systemic strategy to identify lncRNA signatures related to areca nut-induced HNC. 

This study revealed several potentially important points (Figure 7). (1) A total of 28 lncRNAs induced by the areca nut were identified. These included 17 upregulated and 11 downregulated lncRNAs. (2) A total of 45 lncRNAs associated with HNC were identified, including 39 overexpressed and 6 underexpressed lncRNAs in the tumor tissues. (3) A panel of 10 lncRNA signatures associated with areca nut-induced HNC was defined, including eight oncogenic and two tumor-suppressive signatures. (4) Three hub lncRNAs (LUCAT1, UCA1, and MIR31HG) participate in multiple malignant pathways and are associated with poor prognosis of HNC. (5) MIR31HG facilitates cancer progression. Silencing of this molecule attenuates arecoline-induced cell invasion in HNC cells. Thus, this study defined a panel of lncRNA signatures that can be used as biomarkers for areca nut-induced HNC.

Recent high-throughput genomic sequencing techniques have generated abundant library data related to transcriptomic information, including lncRNA structure and expression in biological organisms. TCGA data are comprehensive bioinformatics tools that annotate molecular aberrations at various levels, such as epigenetics, DNA, RNA, and protein, across multiple cancers. Recently, several lncRNA profiles related to HNC have been reported [25,39,40,41,42,43]. However, these studies relied only on database information to highlight candidate molecules. The exclusion of experimental validation may limit insight into the functional pathological significance. PCR arrays lack discovery power or high-throughput capabilities. However, these arrays allow sensitive, specific, and rapid data analysis. Previous studies have reported the use of PCR arrays to analyze genes in specific diseases, such as cancers [33]. In this study, we employed a PCR-based array to analyze 84 cancer-associated lncRNAs in areca nut-treated HNC cells. This approach is an efficient way to address potentially significant lncRNAs without cumbersome genomic searches. After examining 3 HNC cell lines to identify commonly expressed molecules, we identified 28 lncRNAs responding to areca nut induction, including 17 upregulated and 11 downregulated lncRNAs (Figure 1 and Table 1). The upregulated lncRNAs presumably participate in carcinogenic functions, and those that are downregulated may be involved in tumor suppression. 

A prior global survey of lncRNAs in oral cancer patients with areca nut chewing habits compared five pairs of tumors and adjacent normal tissues [34]. An inflammation-related mRNA-lncRNA regulatory network driven by interferon regulatory factors and NF-κB has been identified. However, this study did not directly explore the areca nut-responsive lncRNA and did not further validate the defined molecules. Thus, the associations of the lncRNAs in areca nut-induced carcinogenic mechanisms must be confirmed in future studies. Other relevant studies involving oral fibroblasts showed revealed arecoline induction of three lncRNAs (H19, HIF1A-AS1, and LINC00087), which mediates fibroblastic differentiation or migration [44,45,46]. However, the data were from fibroblasts, limiting their relevance to cancer. To further explore the association of lncRNAs with HNC, we examined the differential expression status of 84 candidate lncRNAs between normal and tumor tissues using TCGA-HNSC data. A total of 45 lncRNAs were differentially expressed in tumor tissues, including 39 overexpressed and 6 underexpressed lncRNAs in the tumors (Figure 2 and Table 2). These lncRNAs represent clinically identified molecules participating in the tumorigenesis of HNC. TCGA-HNSC contains no information related to the areca nut. Thus, we integrated analysis of the areca nut-inducing panel (Figure 1) and TCGA-HNSC tumor panel (Figure 2a) to define a lncRNA signature for areca nut-induced HNC (Figure 3a). A total of ten lncRNAs were identified, including eight upregulated oncogenic and two tumor-suppressive lncRNAs. Six of these lncRNAs were further confirmed to be differentially expressed in the tumors and were induced by arecoline (Figure 3b,c). Recent studies have also reported the involvement of these lncRNAs in HNC, supporting our findings of their roles, including LUCAT1, MIR31HG, UCA1, BLACAT1, NEAT1, and LINC00312 [47,48,49,50,51,52]. Thus, we defined a panel of 10 molecules as the lncRNA signature in areca nut-induced HNC.

Although lncRNAs may modulate gene expression via various mechanisms, the major function of lncRNAs is to process and stabilize mRNA maturation. Thus, lncRNAs are often positively correlated with their downstream effective genes [20,21]. To explore the potential functions of our defined lncRNAs, we employed this concept and used bioinformatics strategies to search for downstream effectors of the lncRNA and predict their regulatory mechanisms in HNC cells (Figure 4a). LUCAT1, UCA1, and MIR31HG were examined. Several positively correlated genes were enriched in various tumorigenic pathways by each specific lncRNA (Figure 4b–d).

LUCAT1 functioned mainly in the modulation of DNA damage repair, stress response, and cell motility in HNC cells (Figure 4b). These results are consistent with those of other studies. In colorectal cancer cells, LUCAT1 increases cell viability by regulating DNA damage-associated genes, resulting in chemoresistance [53]. In lung cancer cells, LUCAT1 (also termed SCAL1) may be upregulated by NRF-2 and may mediate oxidative stress proteins [54]. LUCAT1 promotes cell invasion in several cancer types, including gastric, liver, and oral cancer cells, by modulating YWHAZ, Annexin A2, or PCNA [55,56]. In association with clinical presentations, LUCAT1 was effective in distinguishing between normal and tumor tissues (AUC = 0.768) and was correlated with poor prognosis (HR = 1.368, *p* < 0.05) (Figure 5b,c), suggesting the potential of LUCAT1 as a diagnostic and prognostic marker for HNC. Thus, LUCAT1 may be induced by the areca nut, facilitating cellular stress and cell invasion, in turn leading to a worse prognosis in HNC.

UCA1 promotes malignant progression through various mechanisms in multiple cancers, including HNC [57,58,59,60,61,62,63,64]. In this study, we found the major function of UCA1 in HNC was related to cell motility, including adherent junctions, tight junctions, focal adhesions, and extracellular matrix receptor interactions (Figure 4c). Other studies have described the same findings. In different cancer types, UCA1 may increase cell invasion by modulating Wnt/β-catenin and Notch signaling pathways [61,62], or by sponging specific miRNAs, such as miR-124, miR-126, or miR-143-3p [62,63,64]. Clinically, UCA1 may not serve as a suitable diagnostic marker (AUC = 0.532, Figure 5b). However, the prognostic significance was excellent (HR = 1.640, *p* < 0.05; Figure 5c). Thus, areca nut-induced UCA1 in HNC cells may facilitate cancer invasion and progression, leading to worse treatment outcomes.

MIR31HG modulates carcinogenesis when co-upregulation with miR-31. In lung cancer cells, cigarette smoke condensate may stimulate expressions of MIR31HG and miR-31 to promote cell proliferation [65]. Similarly, the co-upregulation of MIR31HG and miR-31 may increase the proliferation and migration of oral cancer cells [66]. These findings suggest that MIR31HG is a host gene for miR-31. In this study, MIR31HG modulated several malignant functions, including cell proliferation and motility, via various molecular signaling pathways (Figure 4d). These findings are supported by previous observations. MIR31HG may promote cell growth and migration in lung, oral, or thyroid cancer cells [65,66,67], via the C/EBP/Wnt5A or MAP-kinase signaling pathways, depending on the specific cell types [65,67]. We further demonstrated that MIR31HG plays a pivotal role in areca nut-induced cell invasion. Silencing MIR31HG abolished areca nut-induced cell invasion in HNC cells (Figure 6a). This result was validated and supported by the correlative expression of invasion-associated genes, FLNB, LAMA3, and MMP1, in patients with HNC (Figure 6b,c). Thus, the carcinogenic mechanism of areca nut involves the upregulation of MIR31HG, which contributes to cell invasion in HNC. Clinically, MIR31HG was effective in distinguishing between normal and tumor tissues (AUC = 0.682) and correlated with poor prognosis (HR = 1.423, *p* < 0.05) (Figure 5b,c). Thus, MIR31HG may be a diagnostic and prognostic marker for areca nut-induced HNC.

## 5. Conclusions

In conclusion, we defined a lncRNA panel associated with areca nut-induced HNC via integrative analyses of lncRNA profiles related to areca nut induction and dysregulation in patients with HNC. This lncRNA panel consists of ten molecules with eight oncogenic and two tumor-suppressive attributes. The LUCAT1, UCA1, and MIR31HG lncRNAs participate in various malignant functions, which are overexpressed in patients with HNC and are associated with poor prognosis. MIR31HG contributes to areca nut-induced carcinogenesis through the modulation of several invasion-associated genes in HNC. Our defined molecules may be further developed for clinical applications in risk assessment, diagnosis, or prognosis of areca nut-associated malignancy.

## Figures and Tables

**Figure 1 cells-12-00873-f001:**
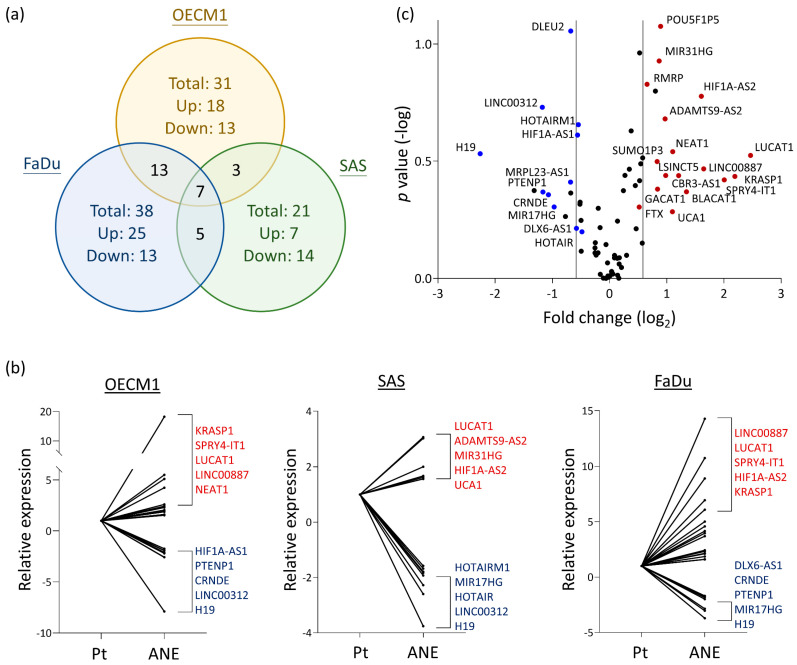
The dysregulated profile of 84 lncRNAs in response to areca nut treatment in three HNC cell lines (OECM1, SAS, and FaDu). (**a**) Summary of the dysregulated lncRNAs in each HNC cell line with chronic treatment by areca nut extract (ANE). (**b**) The most significant dysregulated lncRNAs induced by areca nut (ANE) compared to the parental cells (Pt) in each HNC cell line (OECM1, SAS, and FaDu). (**c**) The volcano plot shows the levels of dysregulated lncRNAs with the average |fold-change| (*X*-axis) and the *p*-values (*Y*-axis) among three HNC cell lines. The red dots represent upregulated lncRNAs (n = 17), and the blue dots represent downregulated ones (n = 11).

**Figure 2 cells-12-00873-f002:**
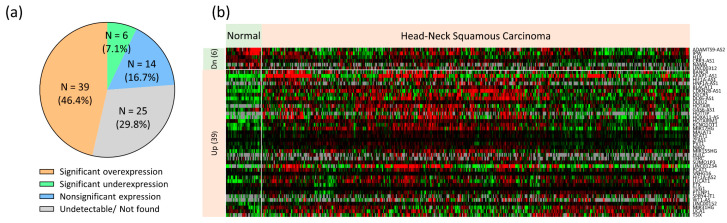
The differential expression profile of 84 lncRNAs in TCGA-HNSC dataset. (**a**) A pie chart of differentially expressed lncRNA in tumors (N = 497) compared to normal tissues (N = 44) as determined by unpaired *t*-test. A total of 45 lncRNAs are altered expression, including 39 overexpressed and 6 underexpressed lncRNAs in tumors. (**b**) Clustergram shows differential expression profiles of the 45 lncRNA across samples. (**c**) Examples of 5 overexpressed lncRNAs in tumors. (**d**) Examples of 5 underexpressed lncRNAs in tumors.

**Figure 3 cells-12-00873-f003:**
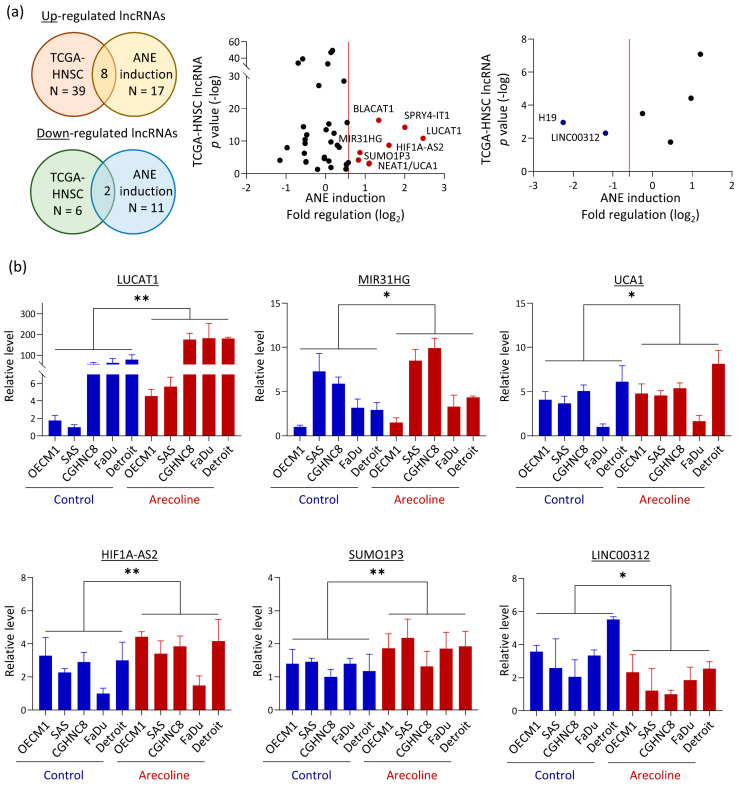
The integration of lncRNAs that were dysregulated in areca nut-treated HNC cells and altered expression in tumors from TCGA-HNSC dataset. (**a**) Overview of eight upregulated lncRNAs and two downregulated lncRNAs, with the levels of fold-change in response to area nut (*X*-axis) and the differential significance (*p* values) between tumors and normal tissues (*Y*-axis). (**b**) Expressions of six lncRNAs (LUCAT1, MIR31HG, UCA1, HIF1A-AS2, SUMO1P3, and LIN00312) dysregulated in five HNC cell lines treated with arecoline (100 μM for 24 h), as examined by RT-qPCR method. (**c**,**d**) Expression of three lncRNAs (LUCAT1, MIR31HG, and UCA1) in three HNC cells treated with arecoline by serial doses (0~200 μM) (**c**) for various time points (0~72 h) (**d**). (**e**) Expressions of six lncRNAs differential presented between 12 HNC cell lines and ten cell lines of normal oral keratinocytes. (*: *p* < 0.05, **: *p* < 0.01, ***: *p* < 0.001).

**Figure 4 cells-12-00873-f004:**
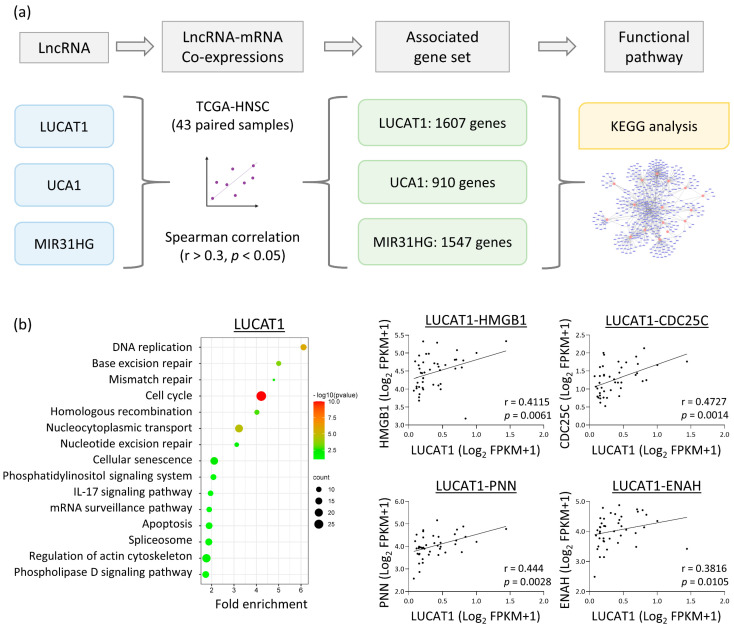
Functional pathways of LUCAT1, UCA1, and MIR31HG, as analyzed via in silico methods. (**a**) The flow chart represents the strategy to investigate the potential functions of lncRNAs (LUCAT1, UCA1, and MIR31HG) through inquiry of the co-expression molecules (mRNAs) and analysis using the DAVID software. (**b**) The LUCAT1 enriched functional pathways and examples of correlative genes. (**c**) The UCA1 enriched functional pathways and examples of correlative genes. (**d**) The MIR31HG enriched functional pathways and examples of correlative genes.

**Figure 5 cells-12-00873-f005:**
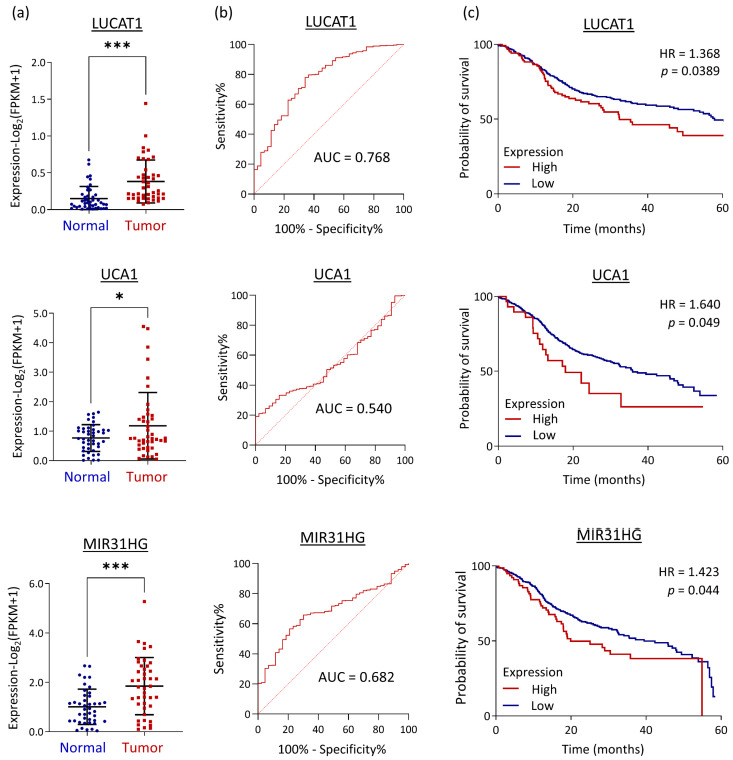
Clinical significance of LUCAT1, UCA1, and MIR31HG in HNC, as assessed by TCGA-HNSC dataset with paired tumor and normal tissues. (**a**) Differential expression levels of LUCAT1, UCA1, and MIR31HG between 43 paired normal and tumor tissues, as determined by paired *t*-test. (**b**) The ROC curves and the AUC values show the diagnostic efficacy of LUCAT1, UCA1, and MIR31HG for HNC using complete TCGA-HNSC dataset. (**c**) Prognostic significance of LUCAT1, UCA1, and MIR31HG for HNC, as demonstrated by the *p*-values and hazard ratios (HRs) via Kaplan-Meier analysis with complete TCGA-HNSC dataset (n = 500). (*: *p* < 0.05, ***: *p* < 0.001).

**Figure 6 cells-12-00873-f006:**
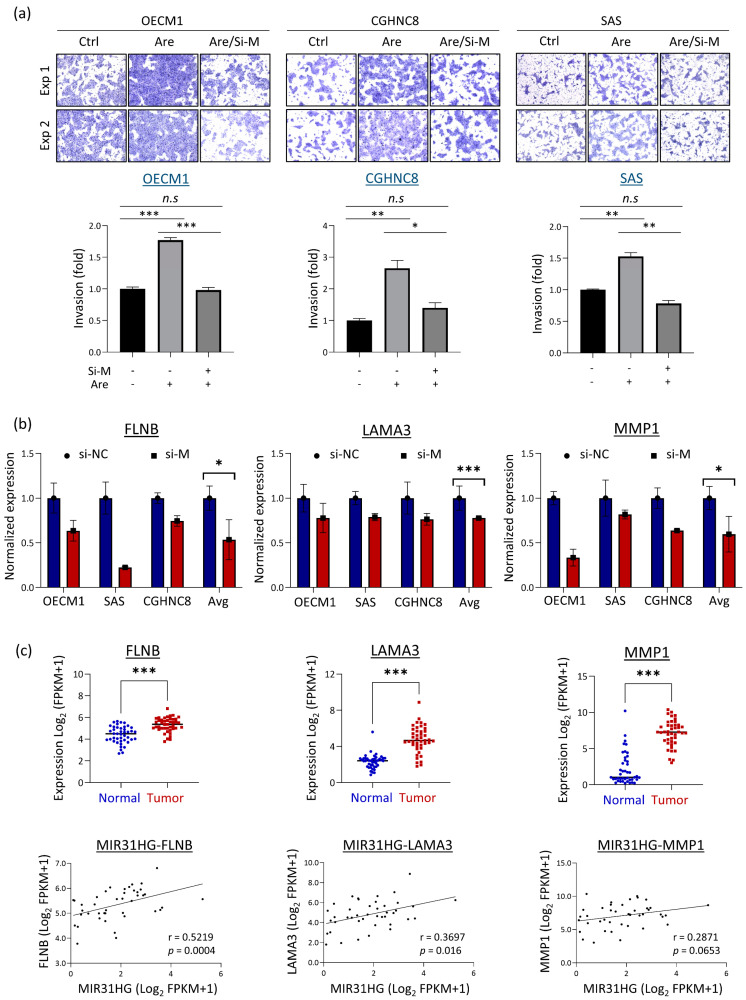
MIR31HG regulates arecoline-induced cellular invasion in HNC cells. The Matrigel invasion assay was performed to analyze the silencing effect of lncRNAs in the regulation of arecoline-induced cell invasion. (**a**) The effect of invasion ability is modulated by MIR31HG silencing. (**b**) The expressions of invasion associated genes (FLNB, LAMA3, and MMP1) in response to MIR31HG silencing in three HNC cell lines, as determined by RT-qPCR analysis. (**c**) The correlative expressions of MIR31HG with three invasion-associated genes. (*: *p* < 0.05, **: *p* < 0.01, ***: *p* < 0.001). si-M, si-MIR31HG; Are, arecoline.

**Figure 7 cells-12-00873-f007:**
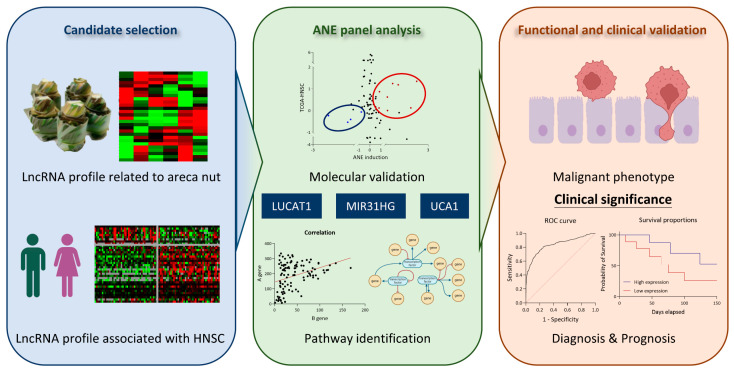
A model of ANE profiling contributes to cell invasion and poor prognosis in HNC.

**Table 1 cells-12-00873-t001:** List of lncRNAs regulated by ANE.

Upregulation				
Symbol	OECM1	FaDu	SAS	Geometric Mean
LUCAT1	5.10	10.74	3.07	5.52
KRASP1	18.25	6.08	−1.16	4.57
SPRY4-IT1	5.50	8.90	1.32	4.01
LINC00887	4.23	14.27	−1.97	3.13
HIF1A-AS2	2.43	6.94	1.67	3.04
BLACAT1	2.30	5.01	1.42	2.54
CBR3-AS1	2.38	3.69	1.40	2.31
NEAT1	2.60	3.96	−1.04	2.15
UCA1	1.32	4.58	1.62	2.14
LSINCT5	1.61	4.15	1.15	1.97
ADAMTS9-AS2	1.18	2.11	3.03	1.96
POU5F1P5	2.04	2.32	1.35	1.85
MIR31HG	1.92	1.59	2.00	1.83
GACAT1	1.88	2.42	1.26	1.79
SUMO1P3	2.36	2.14	1.12	1.78
RMRP	1.36	1.85	1.56	1.57
FTX	1.54	1.60	1.20	1.43
Downregulation				
Symbol	OECM1	FaDu	SAS	Geometric Mean
H19	−7.89	−3.72	−3.76	−4.80
LINC00312	−2.59	−1.71	−2.60	−2.26
PTENP1	−2.20	−2.86	−1.79	−2.24
CRNDE	−2.57	−1.99	−1.80	−2.10
MIR17HG	−1.27	−3.04	−1.93	−1.96
DLEU2	−1.35	−1.81	−1.68	−1.60
MRPL23-AS1	−2.10	−1.24	−1.58	−1.60
DLX6-AS1	−1.93	−1.85	1.07	−1.49
HIF1A-AS1	−2.11	−1.70	1.13	−1.47
HOTAIRM1	−1.74	1.02	−1.83	−1.46
HOTAIR	−1.91	1.60	−2.28	−1.40

**Table 2 cells-12-00873-t002:** Significantly differentially expressed lncRNAs in TCGA-HNSC.

Upregulation				
Entrez gene ID	Gene symbol	Phenotype	Fold change	*p*-value
84740	AFAP1-AS1	Up	24.71	3.77 × 10^−50^
100048912	CDKN2B-AS1	Up	5.94	1.21 × 10^−47^
285987	DLX6-AS1	Up	5.91	1.13 × 10^−39^
8847	DLEU2	Up	2.10	1.55 × 10^−34^
100506465	LINC01234	Up	26.00	1.09 × 10^−33^
10984	KCNQ1OT1	Up	2.96	3.34 × 10^−29^
221883	HOXA11-AS	Up	9.46	8.63 × 10^−28^
101669762	BLACAT1	Up	2.36	3.98 × 10^−17^
114614	MIR155HG	Up	2.44	2.01 × 10^−16^
55000	TUG1	Up	1.20	5.28 × 10^−16^
100750246	HIF1A-AS1	Up	2.53	3.87 × 10^−15^
100642175	SPRY4-IT1	Up	2.93	5.64 × 10^−15^
26220	DGCR5	Up	2.38	3.18 × 10^−14^
5820	PVT1	Up	1.46	3.85 × 10^−13^
100124700	HOTAIR	Up	3.96	1.14 × 10^−12^
100505994	LUCAT1	Up	2.49	1.43 × 10^−11^
100316868	HOTTIP	Up	6.98	3.31 × 10^−11^
100885775	BANCR	Up	2.57	1.77 × 10^−10^
112597	LINC00152	Up	1.68	4.95 × 10^−10^
51352	WT1-AS	Up	6.34	1.88 × 10^−9^
100750247	HIF1A-AS2	Up	1.98	1.91 × 10^−9^
10230	NBR2	Up	1.33	7.91 × 10^−9^
407975	MIR17HG	Up	2.36	1.01 × 10^−8^
554202	MIR31HG	Up	1.68	3.04 × 10^−7^
100506311	HOTAIRM1	Up	1.70	5.46 × 10^−7^
283460	HNF1A-AS1	Up	2.43	8.85 × 10^−6^
9383	TSIX	Up	2.44	2.35 × 10^−5^
474338	SUMO1P3	Up	1.28	6.28 × 10^−5^
11191	PTENP1	Up	1.22	8.14 × 10^−5^
441951	ZFAS1	Up	1.09	1.28 × 10^−4^
728655	HULC	Up	1.87	1.95 × 10^−4^
100507246	SNHG16	Up	1.09	2.65 × 10^−4^
378938	MALAT1	Up	1.14	3.96 × 10^−4^
283131	NEAT1	Up	1.19	6.26 × 10^−4^
652995	UCA1	Up	1.37	9.33 × 10^−4^
100750225	PCAT1	Up	1.36	1.46 × 10^−3^
650669	GAS6-AS1	Up	1.42	1.19 × 10^−2^
100302692	FTX	Up	1.13	4.05 × 10^−2^
7012	TERC	Up	1.31	4.63 × 10^−2^
Downregulation				
Entrez gene ID	Gene symbol	Phenotype	Fold change	*p*-value
100506428	CBR3-AS1	Down	0.61	8.23 × 10^−8^
100507098	ADAMTS9-AS2	Down	0.32	3.75 × 10^−5^
3653	IPW	Down	0.42	3.18 × 10^−4^
283120	H19	Down	0.77	1.11 × 10^−3^
29931	LINC00312	Down	0.72	4.92 × 10^−3^
100996569	NAMA	Down	0.51	1.71 × 10^−2^

**Table 3 cells-12-00873-t003:** ROC analysis of LUCAT1, MIR31HG, and UCA1.

Combination biomarker	*p*-value	AUC (95% CI)	Sensitivity (95% CI)	Specificity (95% CI)
LUCAT1	8.45 × 10^−12^	0.765 (0.689, 0.841)	77.98% (74.12–81.41%)	65.91% (51.14–78.12%)
UCA1	3.98 × 10^−1^	0.532 (0.457, 0.607)	19.18% (15.94–22.90%)	100% (91.70–100%)
MIR31HG	1.90 × 10^−5^	0.682 (0.615, 0.749)	56.60% (52.12–60.98)	79.07% (64.79–88.58%)

## Data Availability

The data of TCGA-HNSC used in this study are from UCSC xena platform (https://xena.ucsc.edu/, accessed on 10 March 2022). The other data presented in this study are available upon request from the corresponding author.

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
