# Peer review of "Molecular Signature of Long Non-Coding RNA Associated with Areca Nut-Induced Head and Neck Cancer"

_cells, 2023, doi:10.3390/cells12060873_

Round 1

Reviewer 2 Report

Dear Editor, 

This is an interesting title and manuscript and authors investigated  systemic strategy to identify lncRNA signatures related to areca nut-induced head and neck cancer. There are some comments which should be followed before next steps;

-It would be better to explain the aim clearly.

- Introduction is poor about head and neck cancer and I suggest you the below references:

-Hajmohammadi E, Ghahremanie S, Alam M, Abbasi K, Mohamadian F, Khayatan D, Rahbar M. Biomarkers and common oral cancers: Clinical trial studies. JBUON. 2021;26(6):2227-37. 

-https://pubmed.ncbi.nlm.nih.gov/34486685/ (https://www.europeanreview.org/article/26522)

-https://pubmed.ncbi.nlm.nih.gov/32135187/ (https://doi.org/10.1016/j.lfs.2020.117483)

-https://www.sciencedirect.com/science/article/abs/pii/S0014299920307494 (https://doi.org/10.1016/j.ejphar.2020.173657)

- Method and materials sections are ok and well categorized.

-Results are ok and understandable 

- It would be better to present the figure 7 in results 

- It would be better to explain more about HNC in first part of discussion and use the above mentioned references.

Best regards,  

Round 2

Reviewer 1 Report

The authors addressed all the concerns and changed the manuscript accordingly. The revised version is significantly better and I recommend that this can be accepted for publication.

Reviewer 2 Report

Dear Editor, 

The revised manuscript is acceptable. 

Best regards,